# Mitochondria-Targeted Liposomes for Drug Delivery to Tumor Mitochondria

**DOI:** 10.3390/pharmaceutics16070950

**Published:** 2024-07-17

**Authors:** Aysegul Ekmekcioglu, Ozgul Gok, Devrim Oz-Arslan, Meryem Sedef Erdal, Yasemin Yagan Uzuner, Meltem Muftuoglu

**Affiliations:** 1Institute of Health Sciences, Department of Medical Biotechnology, Acibadem Mehmet Ali Aydinlar University, 34752 Istanbul, Turkey; aysegul.ekm@gmail.com; 2Faculty of Engineering and Natural Sciences, Department of Biomedical Engineering, Acibadem Mehmet Ali Aydinlar University, 34752 Istanbul, Turkey; ozgul.gok@acibadem.edu.tr; 3School of Medicine, Department of Biophysics, Acibadem Mehmet Ali Aydinlar University, 34752 Istanbul, Turkey; devrim.arslan@acibadem.edu.tr; 4Faculty of Pharmacy, Department of Pharmaceutical Technology, Istanbul University, 34116 Istanbul, Turkey; serdal@istanbul.edu.tr; 5Faculty of Pharmacy, Department of Pharmaceutical Technology, Acibadem Mehmet Ali Aydinlar University, 34752 Istanbul, Turkey; yasemin.uzuner@acibadem.edu.tr; 6Faculty of Engineering and Natural Sciences, Department of Molecular Biology and Genetics, Acibadem Mehmet Ali Aydinlar University, 34752 Istanbul, Turkey

**Keywords:** mitochondria-targeted liposomes, doxorubicin, TPP, drug release, cytotoxicity

## Abstract

The special bilayer structure of mitochondrion is a promising therapeutic target in the diagnosis and treatment of diseases such as cancer and metabolic diseases. Nanocarriers such as liposomes modified with mitochondriotropic moieties can be developed to send therapeutic molecules to mitochondria. In this study, DSPE-PEG-TPP polymer conjugate was synthesized and used to prepare mitochondria-targeted liposomes (TPPLs) to improve the therapeutic index of chemotherapeutic agents functioning in mitochondria and reduce their side effects. Doxorubicin (Dox) loaded-TPPL and non-targeted PEGylated liposomes (PPLs) were prepared and compared based on physicochemical properties, morphology, release profile, cellular uptake, mitochondrial localization, and anticancer effects. All formulations were spherically shaped with appropriate size, dispersity, and zeta potential. The stability of the liposomes was favorable for two months at 4 °C. TPPLs localize to mitochondria, whereas PPLs do not. The empty TPPLs and PPLs were not cytotoxic to HCT116 cells. The release kinetics of Dox-loaded liposomes showed that Dox released from TPPLs was higher at pH 5.6 than at pH 7.4, which indicates a higher accumulation of the released drug in the tumor environment. The half-maximal inhibitory concentration of Dox-loaded TPPLs and PPLs was 1.62-fold and 1.17-fold lower than that of free Dox due to sustained drug release, respectively. The reactive oxygen species level was significantly increased when HCT116 cells were treated with Dox-loaded TPPLs. In conclusion, TPPLs may be promising carriers for targeted drug delivery to tumor mitochondria.

## 1. Introduction

Doxorubicin (Dox) is one of the most widely used effective anthracycline anticancer drugs due to its high efficacy and pronounced antitumoral activity on various types of cancer. On the other hand, the off-target effects of Dox and damage to healthy tissues result in negative side effects of Dox, such as congestive heart failure and bone marrow suppression, and prevent its effective use [1,2]. For this reason, research in the field of reducing the side effects of Dox and increasing its stability and therapeutic efficacy by specifically delivering Dox to its target site, the tumor site, encapsulated in nanocarrier systems has accelerated [3].

Dox exhibits its anticancer effects through various molecular mechanisms. It causes cancer cell death by accumulating DNA damage through the inhibition of Topoisomerase II activity, inhibiting DNA and RNA synthesis by intercalating with DNA, and inducing cell cycle arrest [2]. It also impairs mitochondrial function by causing the production of ROS and disruption of the mitochondrial membrane potential, leading to apoptosis in cancer cells [4]. It has been demonstrated that targeting Dox to tumor mitochondria eliminates the nuclear effects associated with cardiotoxicity [5].

Mitochondria have become a preferential and promising target for many anticancer agents for effective cancer therapy [3,6]. There are different targeting strategies for mitochondria, and triphenylphosphonium (TPP)-based compounds have advantages compared to other approaches to mitochondria targeting. TPP is a lipophilic cation widely used in the preparation of nontoxic nanoparticles targeting mitochondria. It is stable in biological systems, low in chemical reactivity with cellular compartments, a combination of lipophilic and hydrophilic moieties, and obtained through simple synthesis and purification [7,8]. The more negative mitochondrial membrane potential in cancer cells (~−200 mV) than in normal cells (~−140 mV) facilitates the transport of TPP-nanoparticles from the cytoplasm into mitochondria by up to 100-fold compared to non-targeted nanoparticles. This suggests that in cancer cells, the altered mitochondrial membrane potential aids in the uptake of targeted TPP-nanoparticles and eradicates cancer cells through mitochondria-dependent apoptosis [8].

There are many studies targeting Dox to mitochondria with different techniques, and researchers continue to explore and refine various targeting strategies to optimize efficacy and minimize potential drawbacks [9,10,11,12,13,14,15]. Han et al. developed TPP-conjugated Dox and demonstrated that TPP-Dox reverses drug resistance in MDA-MB-435 cancer cells [11]. Chamberlain et al. attached Dox to a mitochondria-penetrating peptide that targets mitochondria and exhibited significant toxicity [9]. Liu et al. conjugated TPP directly to Dox (TPP-Dox) and linked TPP-Dox to hyaluronic acid (HA-ionic TPP-Dox) to form supra-molecular self-assembled structures to achieve mitochondrial targeting. HA-ionic TPP-Dox significantly inhibited tumor growth and prolonged the survival of tumor-bearing zebrafish compared with free Dox [14]. Hou et al. developed TPP-conjugated chitosan nanoparticles loaded with Dox and demonstrated that Dox-loaded TPP nanoparticles specifically disrupted the mitochondria of tumor cells and improved antitumor efficiency in HeLa and A549 cells [12]. Khatun et al. developed Dox-loaded mPEG-TPP conjugates with or without bioreducible disulfide bonds and showed that the bioreducible one can induce fast drug release with enhanced mitochondrial uptake and have a better therapeutic effect than non-bioreducible ones [13]. Cui et al. developed a mitochondria and nucleus dual delivery system that induces elevated levels of apoptosis in Dox-resistant cancer cells [10]. Xiong et al. developed Dox-loaded hyaluronic acid-modified hydroxyapatite nanoparticles (HAP-HA) that activated the mitochondrial apoptotic cascade and increased in vivo anti-tumor efficacy [15].

In the present study, we synthesized and characterized 1,2-distearoyl-*sn*-glycero-3-phosphoethanolamine-N-[amino(polyethylene glycol)-2000] (DSPE-PEG2000-NH_2_)-(3-carboxypropyl)triphenylphosphonium bromide (TPP) (DSPE-PEG-TPP) polymer conjugate and used it to make a nontoxic and pH-dependent mitochondria-targeted Dox-loaded liposomes (TPPLs). Additionally, Dox-loaded 1,2-distearoyl-*sn*-glycero-3-phosphoethanolamine-N-[methoxy(polyethylene glycol)-2000] (18:0 PEG2000 PE) liposomes (PPLs) were prepared and characterized. The physicochemical and cellular properties of empty and Dox-loaded TPPLs were compared with empty and Dox-loaded PPLs and free Dox.

## 2. Materials and Methods

### 2.1. Materials

L-α-Phosphatidylcholine from egg yolk (L-α-PC), cholesterol, (3-carboxypropyl)triphenylphosphonium bromide (TPP), N-hydroxysuccinimide (NHS), N-(3-dimethylaminopropyl)-N′-ethylcarbodiimide hydrochloride (EDCI), triethylamine (TEA), phosphate buffer saline (PBS) tablets, sodium chloride (NaCl), and N-Acetyl cysteine (NAC) were purchased from Sigma-Aldrich (St. Louis, MO, USA). 1,2-Distearoyl-*sn*-glycero-3-phosphoethanolamine-N-[amino(polyethylene glycol)-2000] (ammonium salt) (DSPE-PEG_2000_-NH_2_) and 1,2-distearoyl-*sn*-glycero-3-phosphoethanolamine-N-[methoxy(polyethylene glycol)-2000] (ammonium salt) (18:0 PEG_2000_ PE, PEG-PE) were purchased from Avanti-Polar lipids (Alabaster, AL, USA). Dichloromethane (DCM), ethanol, sodium sulfate (Na_2_SO_4_), sodium acetate trihydrate (CH_3_COONa · 3H_2_O), acetic acid (glacial), and chloroform (CDCl_3_) were purchased from Merck (Taufkirchen, Germany). Lissamine™ Rhodamine B 1,2-dihexadecanoyl-*sn*-glycero-3 phosphoethanolamine, triethylammonium salt (RhB, RhB DHPE), Dulbecco’s phosphate-buffered saline (DPBS), MitoTracker Green FM dye, SlowFade diamond antifade mountant with DAPI, 2′,7′-dichlorodihydrofluorescein diacetate (H_2_DCFDA) and Penicillin/Streptomycin (Pen/Strep) were purchased from Invitrogen-Thermo Scientific (Waltham, MA, USA). The Spectra/Por 6 pre-wetted regenerated cellulose dialysis membrane (MWCO: 2 kDa, 18 mm flat width) was purchased from Spectrum Labs (Miami, FL, USA), and dry regenerated CelluSep cellulose dialysis membrane (MWCO: 10 kDa, 30 mm flat width) was purchased from VWR (Darmstadt, Germany). Tween-80 was purchased from Research Products International (Mt Prospect, IL, USA). Uranyl acetate was purchased from Electron Microscopy Sciences (Hatfield, PA, USA). Dox was purchased from Santa Cruz Biotechnology (Dallas, TX, USA). Fetal bovine serum (FBS) and Dulbecco’s modified eagle medium (DMEM) were purchased from Gibco (Billings, MT, USA). The cell counting kit 8 (CCK-8, WST-8) assay was purchased from Abcam (Cambridge, UK). The HCT116 human colorectal cancer cell line was purchased from the American Type Culture Collection (ATCC, Manassas, VA, USA).

### 2.2. Synthesis and Characterization of DSPE-PEG-TPP Polymer

To prepare mitochondria-targeted liposomes, a DSPE-PEG-TPP polymer conjugate was synthesized as described previously [16]. The mixture of TPP (0.48 mmol), NHS (0.144 mmol), and EDCl (0.144 mmol) was dissolved in CDCl_3_ (0.5 mL) and stirred at room temperature (RT) for 2 h under dark conditions. Then, DSPE-PEG_2000_-NH_2_ (0.036 mmol; dissolved in CDCl_3_) and TEA (0.018 mmol) were added to the mixture and stirred at RT for 24 h under nitrogen. The chloroform (CDCl_3_) was evaporated using a rotary evaporator, and the crude product was dissolved in 2 mL of double distilled water (ddH_2_O). A turbid solution was obtained as soon as the crude product was dissolved. The solution was dialyzed using the dialysis membrane (MWCO: 2 kDa) against 1 L ddH_2_O for 24 h at 4 °C in a cold room. During dialysis, ddH_2_O was changed every 2 h for 5 times and left for 24 h at 4 °C. The dialysate was freeze-dried overnight using FreeZone 6 freeze dry system (Labconco, Kansas City, MO, USA), and the purity of the DSPE-PEG-TPP polymer conjugate was characterized by proton nuclear magnetic resonance (^1^H-NMR, Bruker 400 MHz NMR, Berlin, Germany) [16]. The pure DSPE-PEG-TPP polymer was stored as powder at 4 °C for future studies.

### 2.3. Preparation of Liposomes

Both empty and Dox-loaded liposomes were prepared by the thin film hydration method as described previously [16,17,18] with the following modifications: Empty TPPLs and PPLs were prepared by mixing cholesterol: L-α-PC: DSPE-PEG-TPP or PEG-PE in the ratio of 1:0.92:0.08 (mole %). Briefly, all ingredients were dissolved in DCM according to the liposomal formulation, and then lipid film was formed in a rotary evaporator. The excess amount of DCM was removed using a vacuum desiccator. Then, the dried lipid film was hydrated with DPBS (pH 7.4) and allowed for self-assembly overnight at 4 °C. The hydrated product was sonicated for 30 min in an ultrasonic bath. Then, the final liposomal formulation was obtained by filtering once through 0.45 μm and twice through 0.22 μm filters. To monitor empty TPPLs and PPLs in in vitro environments, RhB-TPPLs and RhB-PPLs were prepared using the thin film hydration method by adding the fluorescent tracking dye 0.005 mole % of RhB DHPE to the composition of the liposomes [19].

To prepare Dox-loaded TPPLs and PPLs, Dox stock solution in methanol (10 mM) was added to the same freshly prepared lipid mixture used for the empty liposomes with a final concentration of 100 µM in DCM, and liposomes were prepared by the thin film hydration method as described above [16,17,18]. To separate free (non-incorporated) Dox from the Dox-loaded TPPLs and PPLs, the formulations were ultracentrifuged at 200,000× *g*, 4 °C for 16 h. After ultracentrifugation, the supernatant was discarded (which included non-incorporated/free Dox), and then the pellet was washed with 1 mL of sterile dH_2_O twice. After separating the free Dox, Dox-loaded TPPLs and PPLs were characterized, and the concentration of the Dox encapsulated into the liposomes was determined using a spectral scanning multimode reader (VarioSkan Flash, Thermo Scientific, USA).

### 2.4. Characterization of Liposomes

The particle size, PDI, and zeta potential (surface charge) of liposomes were measured using the dynamic light scattering (DLS) technique (LiteSizer 500; Anton Paar, Austria). The average size, PDI, and zeta potential of liposomes were represented as mean ± standard deviation (STD) of three independent experiments. The size and morphology of the liposomes were determined by using a transmission electron microscope (TEM, Talos L120C, Thermo Scientific, USA). For TEM analysis, 10 µL of liposomes was introduced onto the copper grids (LC300-Cu-25 Lacey/Carbon 300 Mesh), and then negative staining was performed with 2% uranyl acetate.

### 2.5. Stability of Liposomes

For the stability of the empty liposomes, the empty TPPLs and PPLs were incubated at 4 °C, RT, and 37 °C for 8 weeks. The particle sizes of empty TPPLs and PPLs were recorded weekly, and PDI and zeta potentials were recorded every month at 4 °C, RT, and 37 °C.

### 2.6. Encapsulation of Liposomes

The final concentration of the encapsulated Dox in Dox-loaded TPPLs and PPLs was determined by using a spectral scanning multimode reader (VarioSkan Flash, Thermo Scientific, USA). For this purpose, a calibration curve specific to Dox (0.63–40 µM) was created, and the fluorescence intensity of the solution was measured within the spectrum of excitation/emission 470/490–800 nm, where the maximum fluorescence intensity was observed at 590 nm (λ_max_). Then, the final concentration of the encapsulated Dox was calculated according to the equation of a linear line. Both empty TPPLs and PPLs (used as blank) and Dox-loaded TPPLs and PPLs were disrupted with 100% ethanol for complete release of Dox from the liposomal vesicle. Also, the encapsulation efficiency of Dox was determined by the following equation, encapsulation efficiency (%) = (C*_f_*/C*_i_*) × 100, where C*_f_* is the final concentration of Dox calculated and C*_i_* is the initial concentration of Dox added into the liposomal formulation [20].

### 2.7. In Vitro Drug Release

The Dox release profiles were determined using the dialysis method as described previously [21]. Briefly, Dox-loaded TPPLs and PPLs were added to dialysis tubing (MWCO: 10 kDa) and then introduced into a tube filled with either PBS pH 7.4 with 0.01% Tween-80 or acetate buffer pH 5.6 with 0.01% Tween-80 in a ratio of 1: 80 (liposome: buffer). The tubes were incubated at 37 °C with a constant stirring speed of 100 rpm. A sample from the receptor solution was collected at predetermined time points (0.25, 0.5, 0.75, 1, 2, 3, 4, 5, 24, 48, 72, and 96 h), and the collected volume was replaced with the same amount of fresh buffer. The Dox content released into the receptor buffer was determined by a spectral scanning multimode reader (VarioSkan Flash, Thermo Scientific, USA) and using the calibration curve constructed, where the fluorescence intensity of the standard solutions (0.02–2 μM) was measured within the spectrum of excitation/emission 470/490–800 nm. The maximum fluorescence intensity was observed at 595 nm (λ_max_). Then, the released amount of Dox was calculated from the calibration curve constructed in both pH 7.4 and pH 5.6 buffer with 0.01% Tween-80.

### 2.8. Cellular and Mitochondrial Uptake of Liposomes

The quantification of the cellular uptake of free Dox, Dox-loaded TPPLs, and Dox-loaded PPLs was performed by flow cytometry analysis [16,17,18]. For this purpose, free Dox and Dox-loaded liposomes were added to HCT116 cells at a final concentration of 5 µM of Dox and then incubated for 1 h at 37 °C and 5% CO_2_ incubator. Then, the quantification of the uptake of liposomes into the cell was performed via flow cytometry (BD FACS Verse System, Canton, MA, USA) on PE-A channel [16,17,18]. HCT116 cells that were not incubated with liposomes were used as a control.

The localization of RhB-TPPLs, RhB-PPLs, Dox-loaded TPPLs, and Dox-loaded PPLs in the mitochondria and the cell was visualized with a confocal microscope (Carl Zeiss, Oberkochen, Germany) using the MitoTracker Green FM dye after incubating with HCT116 cells at 37 °C and 5% CO_2_ incubator for 4, 8, and 24 h with the liposomes. The red color of RhB and Dox with the green color of MitoTracker Green was displayed as orange-yellow for liposomes localized in mitochondria. The nucleus was visualized by using SlowFade diamond antifade mountant with DAPI as a fluorescent dye. To detect the fluorescence of RhB, MitoTracker Green, and DAPI, lasers at 560, 488, and 405 nm were used, respectively [16,22].

### 2.9. In Vitro Cytotoxicity of Empty and Dox-Loaded Liposomes

For the cytotoxicity of empty TPPLs and PPLs, impedance-based real-time detection of cell proliferation and cytotoxicity experiments were performed according to the instruction manual of the xCELLigence Real-Time Cell Analysis (RTCA) DP (dual purpose) instrument (Acea Biosciences, Santa Clara, CA, USA). Approximately 22 h after seeding 12,000 cells/well in E-plate 16, when the cells were in the log growth phase, the cells were treated with increasing concentrations of empty TPPLs and PPLs and monitored every 30 min for 96 h. The cells were also treated with a final concentration of 0.01% PBS, which served as a vehicle control. The results were expressed by normalized CI, which was derived from the ratio of CIs before and after the addition of molecules [23].

The cytotoxic effects of free Dox and Dox-loaded TPPLs and Dox-loaded PPLs were measured in HCT116 cells using WST-8 assay according to the manufacturer’s instructions. Briefly, 5 × 10^3^ HCT116 cells per well were seeded in 96-well plates and incubated overnight at 37 °C, 5% CO_2_ incubator. Then, the cells were treated with different concentrations of free Dox, empty TPPLs and PPLs, Dox-loaded TPPLs, and Dox-loaded PPLs and incubated for 48 h at 37 °C, 5% CO_2_ incubator. Subsequently, WST-8 reagent was added into each well, and the absorbance of each well was measured at 460 nm using a plate reader (Powerwave XS2, BioTek, Winooski, VT, USA). The percent cell survival was calculated by dividing the absorbance (OD 460 nm) of treated cells by the absorbance of untreated cells and then multiplied by 100. All experiments were performed in triplicates as three independent experiments. The control cells for the normalization were the non-treated, empty TPPLs and empty PPLs cells. The IC_50_ values were calculated using GraphPad Prism 9.3.1 software.

### 2.10. Reactive Oxygen Species (ROS) Measurements

A cellular ROS indicator H_2_DCFDA was used to measure cellular ROS according to the manufacturer’s instructions. Briefly, 2 × 10^4^ HCT116 cells per well were plated in clear bottom black-walled 96-well plates. Following overnight incubation at 37 °C, 5% CO_2_ incubator, the cells were treated with 5 μM free Dox, Dox-loaded TPPLs, and Dox-loaded PPLs alone or in combination with the ROS scavenger 1 mg/mL NAC for 24 h. Then, the cells were incubated with 10 μM H_2_DCFDA for 45 min at 37 °C, 5% CO_2_ incubator. ROS levels were measured at excitation/emission 485/535 nm using a multimode microplate reader (VarioSkan Flash, ThermoFisher Scientific, Waltham, MA, USA). The experiment was repeated three times. Fluorescence data were normalized to the corresponding cell viability data, which were measured using the WST-8 assay according to the manufacturer’s instructions [23].

### 2.11. Statistical Analysis

All quantitative data were presented as the mean ± STD of three experiments. Statistical analysis was performed using a two-tailed Student’s *t*-test for pairwise comparison and one-way ANOVA with multiple comparisons by Tukey post hoc tests for multiple comparisons. Statistical analysis was performed using the GraphPad Prism 9.2.0 software. *p* < 0.05 was considered statistically significant.

## 3. Results

### 3.1. Synthesis and Characterization of DSPE-PEG-TPP Polymer Conjugate

TPP was conjugated to DSPE-PEG, which is a very useful material for the formulation of liposomes for achieving prolonged blood circulation time, improved stability, and enhanced encapsulation efficiency [24]. Figure 1 shows that the TPP-conjugated amphiphilic DSPE-PEG polymer has been successfully synthesized and obtained in pure form. The proton NMR analysis was conducted in CDCl_3_ as the solvent, with a peak at 7.24 ppm. The reference peak seen at 0 ppm is the tetramethyl silane (TMS) signal, which is the internal standard substance in the solvent. The multiplet seen in the spectrum around 7.2–7.4 ppm comes from the aromatic protons of the TPP molecule. The peak around 8 ppm belongs to the amide proton in the molecule, and since this proton is labile, the integral value was not consistent, which is an expected situation in CDCl_3_. The –OCH_2_ protons in the ethylene glycol monomers of the DSPE-PEG chain appeared at a broad peak around 3.6 ppm, as expected [25]. The aliphatic CH_2_ protons of polyethylene (PE) chains appeared at 1.23 ppm. The two ester protons (–C=OOCH_2_) at the end of the DSPE-PEG chain and the four ester protons in the PE chains appeared in the 4.4–3.8 ppm range. Based on their integration ratio, it can be understood that the reaction occurred at a one-to-one ratio of TPP molecule and DSPE-PEG polymer. The integration of DSPE-PEG ester protons, especially their retention with the tertiary 1 proton in the PE polymer, indicates that the polymer preserves its chemical structural integrity. Moreover, the integration of this tertiary proton peak, as the value of 1, is consistent with the aromatic 15 protons of the TPP molecule, proving that there is no TPP-derived impurity left in the product.

### 3.2. Characterization of Empty and Dox-Loaded Liposomes

The characterization results for liposomes are presented in Table 1. The size of all liposomes is under 200 nm. The zeta potential measurement demonstrates the surface charge of liposomes, where the PPLs, Dox-PPLs, and RhB-PPLs are negative due to PEG-PE, and that of TPPLs, Dox-TPPLs, and RhB-TPPLs is positive due to DSPE-PEG-TPP (Table 1). Dox is a positively charged molecule due to the presence of an amino group. PPL has a negative surface charge. When Dox is encapsulated in a PPL, the positively charged Dox molecules adsorb onto the negatively charged surface of the PPL. This can partially neutralize the negative surface charge and lead to a more negative zeta potential of the Dox-loaded PPL compared to the empty PPL (Table 1). Both Dox and DSPE-PEG-TPP groups carry a positive charge. When Dox is encapsulated in a TPPL, the positively charged Dox molecules adsorb onto the positively charged surface of the TPPL. The positive charges from both Dox and TPP accumulate on the liposome surface, leading to an overall more positive zeta potential for the Dox-loaded TPPL compared to the empty TPPL. The positive charge from the TPP groups alone in an empty TPPL determines the zeta potential, which is less positive than that of the Dox-loaded TPPL (Table 1). The values given in Table 1 show the mean ± STD of three independent experiments.

TEM images of empty and Dox-loaded TPPLs and Dox-loaded PPLs are presented in Figure 2. TEM results demonstrate that the liposomes were structurally spherical, the core parts of Dox-loaded ones were dark black (filled), and the core parts of empty liposomes were light gray (empty) (Figure 2).

### 3.3. Stability of Empty PPLs and TPPLs

For the stability analysis of empty TPPLs and PPLs, the time- and temperature-dependent changes (4 °C, RT, and 37 °C) in their sizes, PDI values, and zeta potentials were measured weekly and followed for 8 weeks (Figure 3). No statistically significant changes were observed in the size of TPPLs and PPLs over 8 weeks at 4 °C and RT (Figure 3A,B). There were no significant changes in the size of PPLs for 8 weeks at 37 °C, whereas fluctuations were observed in the size of TPPLs (Figure 3C). The PDI value of PPLs did not change statistically significantly at all temperatures for 8 weeks, but it was more stable at 4 °C (Figure 3D). The PDI value of TPPLs did not change significantly and was around 0.2 at 4 °C and RT for 8 weeks, but increasing RT to 37 °C caused an increment in the PDI values of TPPLs (Figure 3E). The zeta potentials for PPLs were negative at all temperatures for 8 weeks (Figure 3F), whereas TPPLs lost their positive surface charge and became negative after a month at 37 °C (from 5.77 ± 0.15 to −5.00 ± 0.10) (Figure 3F). The zeta potentials for TPPLs were positive at RT for 8 weeks but decreased from 6.07 ± 0.32 to 2.00 ± 0.17 after a month (Figure 3F). It can be concluded that considering all parameters, both PPLs and TPPLs are more stable at 4 °C (Figure 3).

### 3.4. Determination of Encapsulation Efficiency

The amount of the encapsulated drug in the liposomes was determined by using a spectral scanning multimode reader (VarioSkan Flash Plate Reader, Thermo Scientific). The fluorescence intensity of the formulation was measured at 590 nm (λ_max_), which was the wavelength at which Dox gave the maximum fluorescence intensity after the disruption of liposomal vesicles using 100% ethanol. And then, by using the calibration curve specific to Dox, the concentration of the drug encapsulated in the liposomal formulation was calculated. Dox-loaded liposomes showed characteristics similar to those of empty liposomes through both size and charge distribution (Table 1). According to the calculations, the encapsulation efficiency of Dox in PPLs and TPPLs was found to be 23.79% and 11.13%, respectively. Although encapsulation efficiency was low, it was sufficient for in vitro analysis in this study. The encapsulation efficiency of PPLs was higher than that of TPPL. This situation might be caused by the steric hindrance between the molecules. The encapsulation efficiency of TPPLs might be decreased since the DSPE-PEG-TPP polymer conjugate is larger than PEG-PE.

### 3.5. Drug Release

Different concentrations of Dox, ranging from 0.02 to 2 µM, were used for the construction of the calibration curve and plotted the mean absorbance value versus the concentration of Dox. The Dox calibration curve was recorded at the wavelength of 595 nm (λ_max_). The linear range of the calibration curve obtained for the Dox solution showed good linearity, and the R square of the regression equation was 0.999. The amount of Dox released from TPPLs increased statistically significantly at pH 5.6 compared to pH 7.4 (Figure 4A). In 1 h incubation, 10.09 ± 1.08% of Dox was released from TPPLs at pH 7.4, whereas 20.23 ± 1.90% Dox was released from TPPLs at pH 5.6 (Figure 4A, the embedded figure focused on the first 4 h). During the first 5 h, burst release was observed, with 34.41 ± 1.75% at pH 7.4 and 57.97 ± 1.20% at pH 5.6 from TPPLs. Then, the sustained release was observed, and after 72 h, TPPLs released almost all the encapsulated Dox at pH 5.6 (96.01 ± 1.35%), whereas 54.79 ± 2.91% of Dox was released at pH 7.4 (Figure 4A). There was no statistically significant difference in the amount of Dox released from PPLs at pH 7.4 and 5.6 until the 72 h time point. At acidic pH 5.6, 99.60 ± 0.21% Dox was released from PPLs at 96 h, whereas 73.50 ± 0.87% Dox was released at physiological pH 7.4 (Figure 4B).

### 3.6. Cellular Uptake and Cytotoxicity of Empty PPLs and TPPLs

To track empty TPPLs and PPLs in in vitro environments, RhB-TPPLs and RhB-PPLs were prepared and characterized (Table 1). The localization of RhB-TPPLs in the mitochondria was visualized by a confocal microscope using MitoTracker Green after incubating with HCT116 cells for 4, 8, and 24 h. The red color of RhB and the green color of MitoTracker Green are displayed as yellow/light orange for liposomes localized in mitochondria. The nucleus was stained with DAPI [16,22]. Figure 5A shows that RhB-PPLs, which do not have a mitochondria targeting group, did not localize to mitochondria at all the time points tested (Figure 5B, merged green dots). In Figure 5B, RhB-TPPLs with the mitochondria targeting group TPP-PEG-PE started to localize in the mitochondria from the 4th hour, and their localization increased at the 8th hour and then the 24th hour (Figure 5B, merged, yellow/light orange dots).

The effect of empty TPPLs and PPLs on the viability of HCT116 cells was determined by the xCELLigence RTCA DP system. TPPLs and PPLs were applied to the cells at five different concentrations and did not show any cytotoxic effect on the cells even at the highest concentration applied (Figure 5C).

### 3.7. Cellular Uptake of Dox-Loaded Liposomes

The cellular uptake capacity of Dox-loaded TPPLs and PPLs was quantified by flow cytometry analysis [3]. When the populations of unstained and treated cells were compared, a shift was observed, indicating that Dox-loaded liposomes were internalized within the cells. According to the fluorescence intensities obtained, Dox uptakes from PPLs and TPPLs were increased by 1.36 ± 0.0-fold and 1.51 ± 0.14-fold compared to free Dox upon 1 h of incubation, respectively (Figure 6A).

Intracellular delivery of Dox-loaded TPPLs and PPLs was also confirmed via confocal microscopy imaging. The overlap of the red color of Dox and the green color of MitoTracker Green is displayed as yellow/light orange for liposomes localized in mitochondria. The nucleus was stained with DAPI. Dox-loaded TPPLs were localized in the mitochondria starting from the first time point (4th hour) (Figure 6B), whereas Dox-loaded PPLs did not localize in mitochondria (Figure 6C). The intensity of localization of Dox-loaded TPPLs was increased as incubation time increased from the 8th to the 24th hour. (Figure 6C). Thus Dox-loaded TPPLs accumulated in mitochondria in a time-dependent manner.

### 3.8. Cytotoxicity of Dox-Loaded TPPLs and PPLs

The dose-dependent cytotoxicity of free Dox, Dox-loaded TPPLs, and Dox-loaded PPLs in HCT116 cells is shown in Figure 7. After 48 h incubation, IC_50_ values of free Dox, Dox-loaded TPPLs, and Dox-loaded PPLs were 0.55 μM (Figure 7A), 0.34 μM (Figure 7B), and 0.47 μM (Figure 7C), respectively. The IC_50_ value of Dox-loaded TPPLs was 1.62-fold lower than that of free Dox and 1.42-fold lower than that of Dox-loaded PPLs (Figure 7). The IC_50_ value of Dox-loaded PPLs was also 1.17-fold lower than that of free Dox.

### 3.9. ROS Levels

The intracellular ROS levels were measured using H_2_DCFDA, which is oxidized to highly fluorescent 2′,7′-dichlorofluorescein (DCF) in the presence of ROS. The fluorescence data was normalized to the corresponding cell viability data at 24 h Dox incubation because of the observed cell death at the 24 h post-treatment point. Dox-loaded TPPLs showed a statistically significantly increased amount of ROS production in HCT116 cells compared to free Dox and Dox-loaded PPLs (Figure 8). The treatment of cells with ROS scavenger NAC together with Dox did not cause a change in ROS levels, indicating that an increase in ROS in the cells was due to Dox treatment (Figure 8).

## 4. Discussion

Anticancer drugs known to be effective on intracellular organelles become more advantageous when encapsulated in a nanocarrier system and delivered to the target organelle [18,26,27,28,29]. Despite advancements in drug delivery technologies, further research is needed to improve the targeting performance and safety of drug delivery systems. There are several examples in the literature regarding the mitochondria-targeting moieties incorporated into the nanoparticles [9,10,11,12,13,14,15]. However, the use of the TPP group conjugated with DSPE-PEG chains on the surface has not yet been displayed for self-assembled liposomal formulations in terms of their stability and pH dependency. In this study, the mitochondria-targeting polymer, DSPE-PEG-TPP polymer conjugate, was successfully synthesized, and Dox molecules were encapsulated into these stabilized, mitochondria-targeted liposomes to enhance the therapeutic index of Dox. Then, a comparative study was conducted to characterize the physicochemical properties and cytotoxicity of both empty and Dox-loaded TPPLs and PPLs.

All liposomal formulations prepared in this study were smaller than 200 nm in size. This suggests that they have a longer circulation time and accumulate more effectively in the tumor microenvironment due to the enhanced permeability and retention (EPR) effect [30]. The PDI of all liposomes is below 0.2, indicating that the size distribution of the liposomes is monodisperse [31]. The zeta potential of mitochondria-targeted liposomes was positive due to the DSPE-PEG-TPP polymer conjugate, whereas non-targeted PEGylated liposomes were negatively charged. According to characterization based on the physical structure of liposomes, both empty and Dox-loaded liposomes were spherically shaped. However, since TEM images were taken under a high vacuum, water in the aqueous core of the liposomes may evaporate, and thus liposomes may shrink as the water evaporates. Therefore, the size of liposomes obtained in TEM images may be smaller than determined in DLS, and a slightly distorted round image may be obtained in TEM [32,33].

The stability of the liposomes depends both on chemical factors, such as oxidation of the lipid content and the formation of free radicals in fatty acid tails, and physical factors, such as storage temperature and environment [34]. Thus, the stability of the liposomes was determined based on the storage temperature. According to the physicochemical properties, both TPPLs and PPLs were stable at 4 °C. Similarly, Muppidi et al. demonstrated that the stability of PEGylated liposomes based on their size was favorably stable at RT and 4 °C for 3 months but not at 37 °C. Moreover, they observed fungal growth in liposomes stored at 37 °C and RT after 1 month and 3 months, respectively. Therefore, the only stable temperature condition for liposomes was considered to be 4 °C [35]. In another study, the PEGylated liposomes stored at 4 °C were stable for more than 150 days compared to other storage conditions, which were −20 °C and RT [36]. In this study, cloudiness was seen only in the liposomes stored at 37 °C after 8 weeks. Considering the size, PDI and zeta potential parameters of both PPLs and TPPLs are more stable for 8 weeks at 4 °C.

Since the pH of the tumor tissue is acidic (pH 5.6–6.8) [14,27], the in vitro release profiles of Dox-loaded TPPLs and PPLs were analyzed at two different pH levels—physiological pH and acidic pH. The concentration of Dox released from both TPPLs and PPLs was higher at acidic pH 5.6, which indicates that drug release occurs in the tumor microenvironment. Although both TPPLs and PPLs showed an increased release rate at pH 5.6 compared to pH 7.4, the release rate from TPPLs was higher than that from PPLs during the first 5 h. Hou et al. also reported that there was a pH-dependent drug release profile for Dox. The release rate of Dox from nanoparticles is enhanced as the pH of the medium decreases [12].

The cellular uptake capacity of the fluorescently (RhB) labeled liposomes was demonstrated by confocal microscopy. According to the result obtained after mitochondrial and nuclear staining, it was observed that DSPE-PEG-TPP-conjugated RhB-labeled liposomes were specifically and effectively localized in the mitochondria, whereas non-targeted liposomes were not. Additionally, as the incubation time increased, the intensity of the co-localization in mitochondria was also increased. Even so, the localization time of liposomes varied from study to study since different types of cells, liposomal formulations, and time points were chosen during investigations [16,18,37]. There was an intercellular accumulation of mitochondria-targeted Rh-TPP-DSPE-PEG and Rh-Dequalinium-DSPE-PEG after 1 h of incubation in B16F10 murine melanoma cells [37]. In another study, the internalization into A549 cells and localization in mitochondria of TPGS_1000_-TPP liposomes were observed within 30 min and 24 h, respectively [18].

The empty TPPLs and PPLs were not toxic to HCT116 cells. Similarly, Biswas and colleagues demonstrated that mitochondria-targeted TPP-PEG-L-8% was nontoxic to HeLa cells after 24 h incubation. However, a mitochondria-targeted liposome with stearyl residue, STPP-L-8%, was found to be toxic starting from 15.63 μg/mL concentration. The toxicity difference between the liposomes was explained by the toxicity of the stearyl moiety [16]. Zhou et al. demonstrated that empty TPGS_1000_-TPP liposomes were toxic to A549 cells at concentrations higher than 1 μM and suggested that this cytotoxicity was caused by TPGS_1000_ [18]. Moreover, Kang et al. reported that empty TPP-DSPE-PEG and DQA-DSPE-PEG liposomes were not cytotoxic to B16F10 murine melanoma cells. TPP- or DQA-conjugated liposomes carrying resveratrol showed no statistically significant difference in terms of accumulation of resveratrol in mitochondria, anticancer activity, ROS production, and mitochondrial depolarization in B16F10 cells [37]. Li and colleagues also demonstrated that empty TPP-PEG liposomes were nontoxic to MDA-MB-231 cells [38].

Delocalized lipophilic cations (DLCs), including TPP, Rh-123, F16, and dequalinium (DQA), easily penetrate mitochondria due to the negative transmembrane potentials of mitochondria. Their high accumulation can lead to toxicity in the mitochondria, with different toxicity mechanisms. For example, Rh-123 inhibits ATP synthase activity, and F16 induces mitochondrial permeability transition pore opening. It has been demonstrated that TPP-based compounds in mitochondria are in dynamic equilibrium and can be safely administered in high doses in the long term without causing serious damage to organs such as the heart, liver, and kidney [39]. The limitations of DQAsomes, including low transfection efficiency, short half-life, and potential toxicity concerns, hinder their broader application [40]. Although Dox-loaded DQAsomes were shown to have less surface charge [41], TPP-conjugated liposomes prepared in this study provided even lower yet still positive zeta potential. It is well known that high positive zeta potential would increase the elimination of the reticuloendothelial system and induce nonspecific toxicity [41].

In conclusion, various studies have demonstrated that the mitochondria-targeting group, TPP, either by itself or when conjugated with PEG-PE, has no cytotoxic effect on different types of cancer cells [16,18,37,38]. All Dox-loaded liposomal formulations exhibited higher cytotoxicity in comparison with free Dox. In addition to this, Dox-loaded TPPLs showed a lower IC_50_ value compared to free Dox and Dox-loaded PPLs. This could be attributed to the efficient cellular uptake of Dox-loaded TPPLs by cancer cells. Furthermore, ROS produced by the Dox-loaded TPPLs was statistically significantly higher compared to that produced by free Dox and Dox-loaded PPLs. Hou et al. also demonstrated that ROS generated by TPP-Dox nanoparticles was higher than that generated by free Dox in A549 and HeLa cells at each time point (12, 24, 48 h) [12]. This study demonstrated that TPPLs are a promising vehicle for targeted drug delivery to specifically tumor mitochondria.

## Figures and Tables

**Figure 1 pharmaceutics-16-00950-f001:**
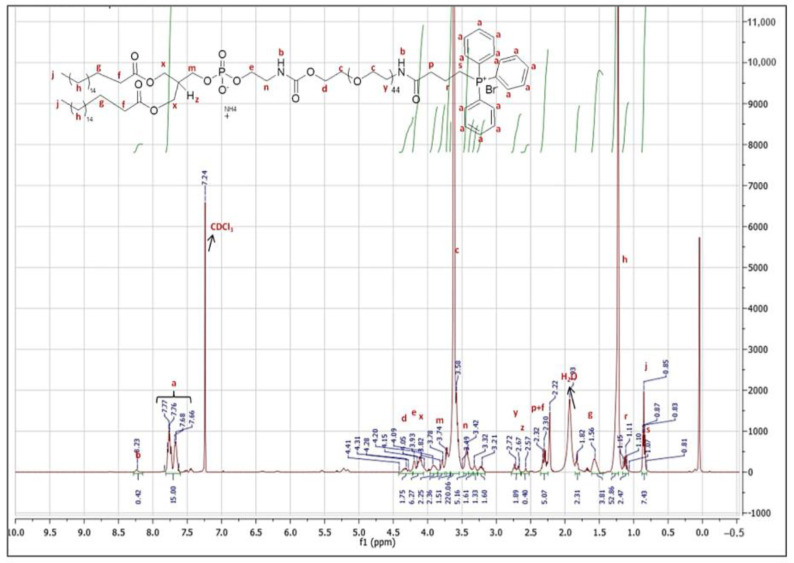
Proton nuclear magnetic resonance (^1^H-NMR) spectrum of DSPE-PEG-TPP polymer conjugate.

**Figure 2 pharmaceutics-16-00950-f002:**
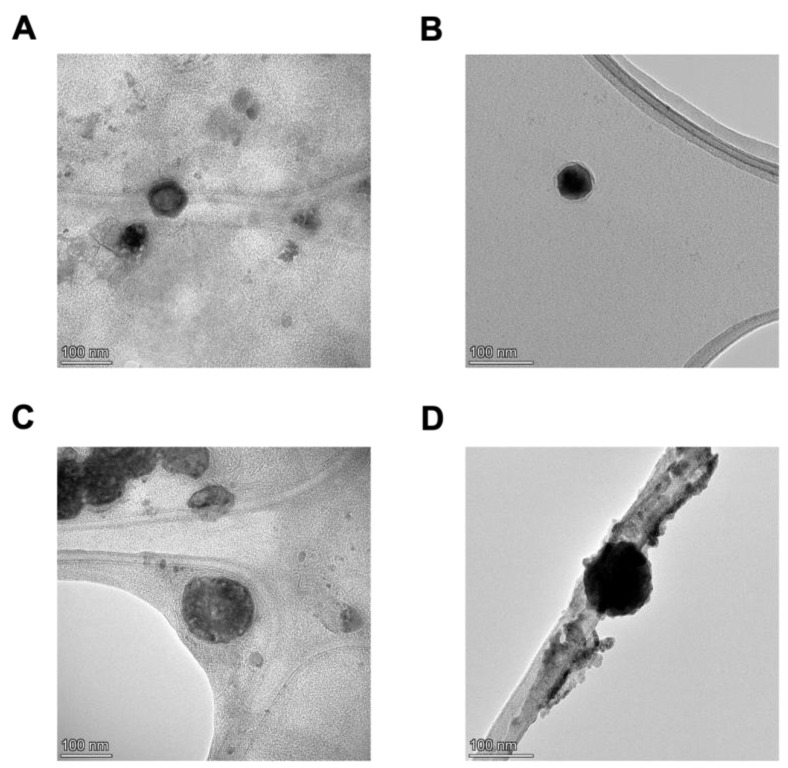
TEM images of PEG-PE and TPP-PEG-PE liposomes. (**A**) Empty PPL (100 nm). (**B**) Dox-loaded PPL (100 nm). (**C**) Empty TPPL (100 nm). (**D**) Dox-loaded TPPL (100 nm).

**Figure 3 pharmaceutics-16-00950-f003:**
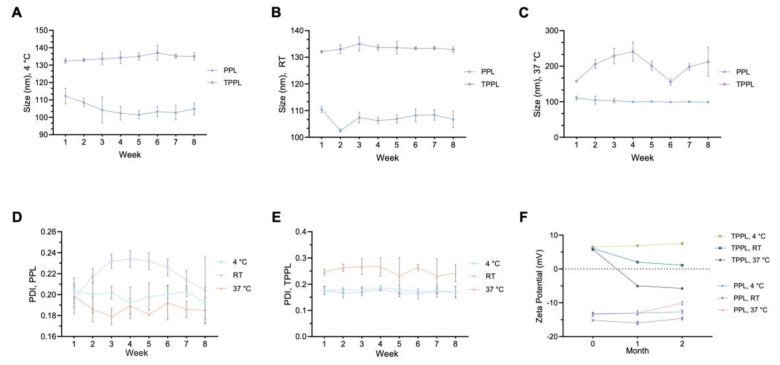
Stability of empty liposomes. Time (1–8 weeks)- and temperature (4 °C, RT and 37 °C)-dependent changes according to (**A**–**C**) size, (**D**,**E**) polydispersity index (PDI), (**F**) zeta potential. The data are represented as the mean ± STD of three independent experiments.

**Figure 4 pharmaceutics-16-00950-f004:**
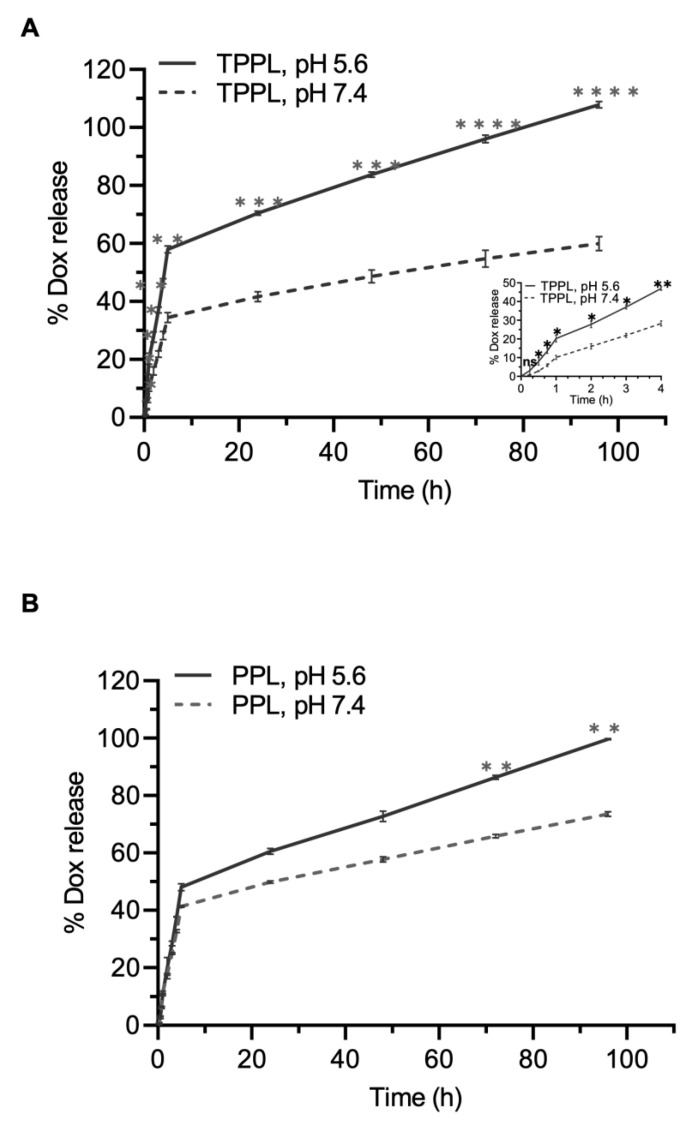
Dox release from (**A**) TPPLs and (**B**) PPLs in pH 7.4 and 5.6 buffers. One-way ANOVA using Tukey’s multiple comparison tests was conducted for each time point; *p* < 0.05 was accepted as statistically significant. Each experiment was performed in triplicate. * *p* = 0.014; ** *p* = 0.003; *** *p* = 0.0008; **** *p* ≤ 0.0001.

**Figure 5 pharmaceutics-16-00950-f005:**
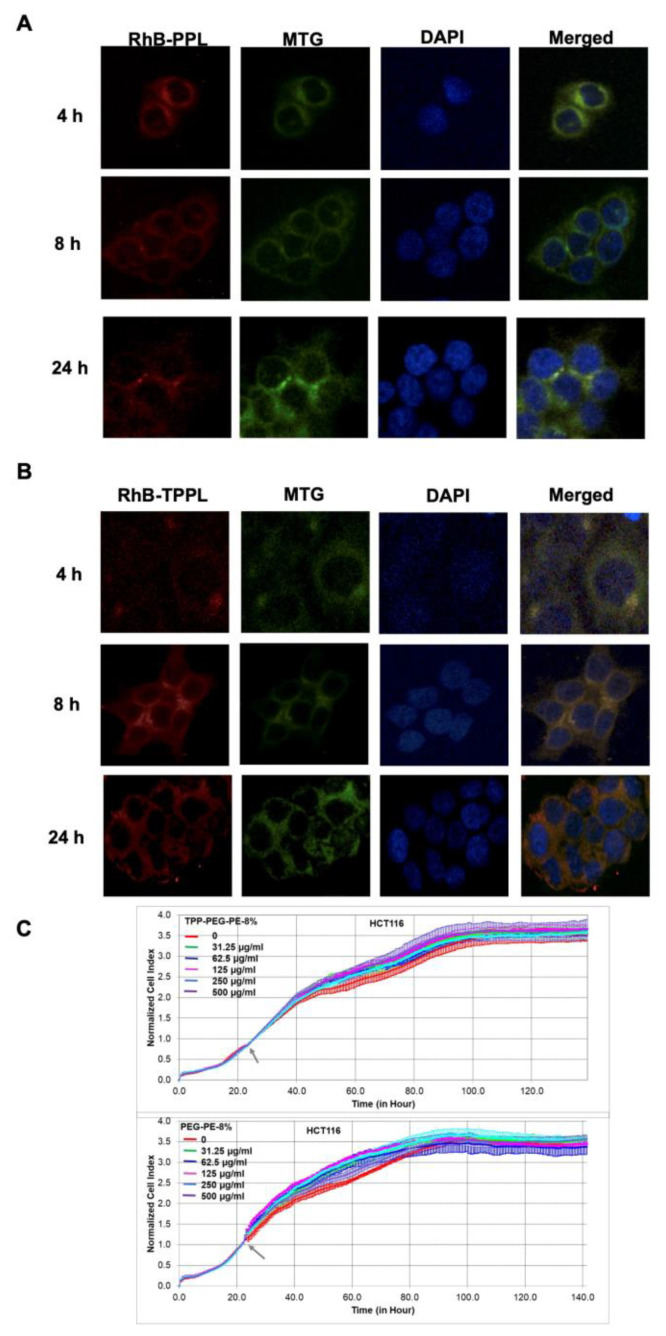
Cellular uptake of fluorescently labeled RhB-PPLs and RhB-TPPLs in HCT116 cells by confocal microscopy. HCT116 cells were incubated with RhB-PPLs (red, 560 nm) and RhB-TPPLs (red, 560 nm) for 4, 8, and 24 h at 37 °C and 5% CO_2_ incubator, and then, mitochondria were stained with the MitoTracker Green FM dye (green, at 488 nm), and nucleus was stained with DAPI (blue, 405 nm). The red color of RhB with the green color of MitoTracker Green was displayed as orange-yellow for liposomes localized in mitochondria in the merged images. (**A**) Localization of RhB-PPLs in HCT116 cells. (**B**) Localization of RhB-TPPLs in the mitochondria of HCT116 cells. (**C**) Real-time dynamic monitoring of the cytotoxic effects of empty TPPLs and PPLs on HCT116 cells using the xCELLigence system. Cell growth was continuously monitored every 30 min. Cell index was normalized to the time point of liposome application. Normalized cell index was plotted as the mean value from triplicates; error bars represent the standard deviation for the mean. The grey arrow indicates the time of empty TPPL and PPL addition.

**Figure 6 pharmaceutics-16-00950-f006:**
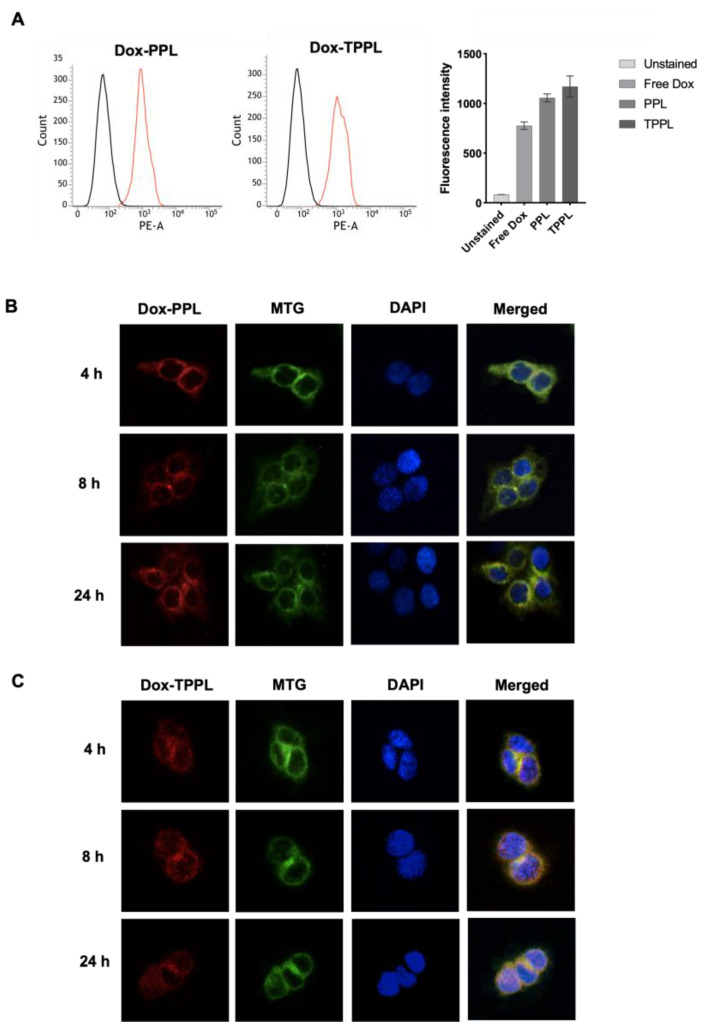
Cellular uptake of Dox-loaded PPLs and TPPLs in HCT116 cells. (**A**) Flow cytometry data for HCT116 cells incubated with Dox-loaded liposomes. HCT116 cells were incubated with 5 μM free Dox and Dox-loaded liposomes for 1 h at 37 °C and 5% CO_2_ incubator. The uptake of the liposomes into the cells was quantified using flow cytometry, measuring the signal on the PE-A channel. HCT116 cells not incubated with any liposomes were used as a control. Black histogram: Control, untreated cells; Red histogram: Dox-loaded liposome treated cells (**B**) Confocal microscopy images of Dox-loaded PPLs. (**C**) Confocal microscopy images of Dox-loaded TPPLs. HCT116 cells were incubated with Dox-loaded TPPLs (red, 560 nm, for 4, 8, and 24 h at 37 °C and 5% CO_2_ incubator, and then mitochondria were stained with the MitoTracker Green FM dye (green, at 488 nm), and nucleus was stained with DAPI (blue, 405 nm). The red color of Dox with the green color of MitoTracker Green was displayed as orange-yellow for liposomes localized in mitochondria in the merged images.

**Figure 7 pharmaceutics-16-00950-f007:**
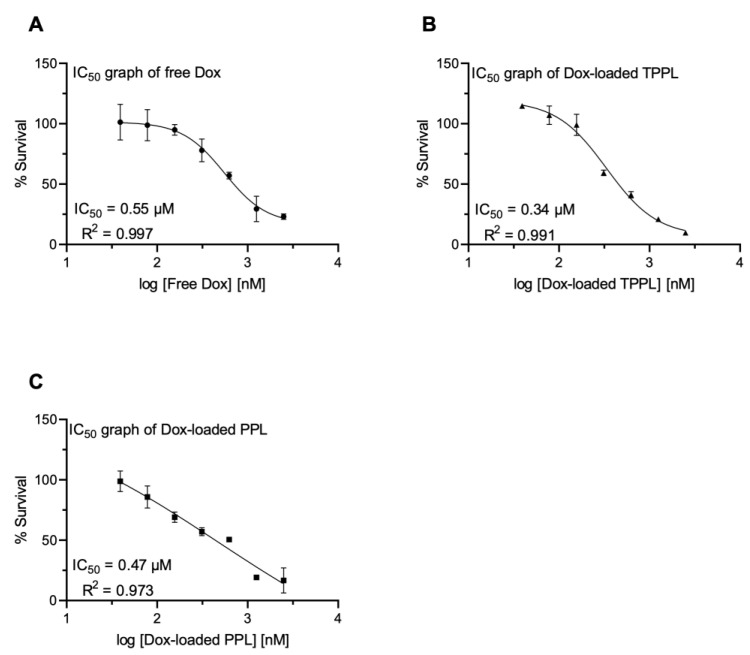
The 50% inhibitory concentrations (IC_50_) of the (**A**) free Dox, (**B**) Dox-loaded TPPLs and (**C**) Dox-loaded PPLs in HCT116 cell lines at 48 h. The percent cell survival was calculated as the mean value (OD 460 nm) of three independent experiments performed in triplicate compared to that of corresponding control cells multiplied by 100. The control cells for the normalization were the non-treated, empty TPPL, and empty PPL cells. The graphs were generated using GraphPad Prism 9.3.1 software for IC_50_ calculations based on WST-8 data.

**Figure 8 pharmaceutics-16-00950-f008:**
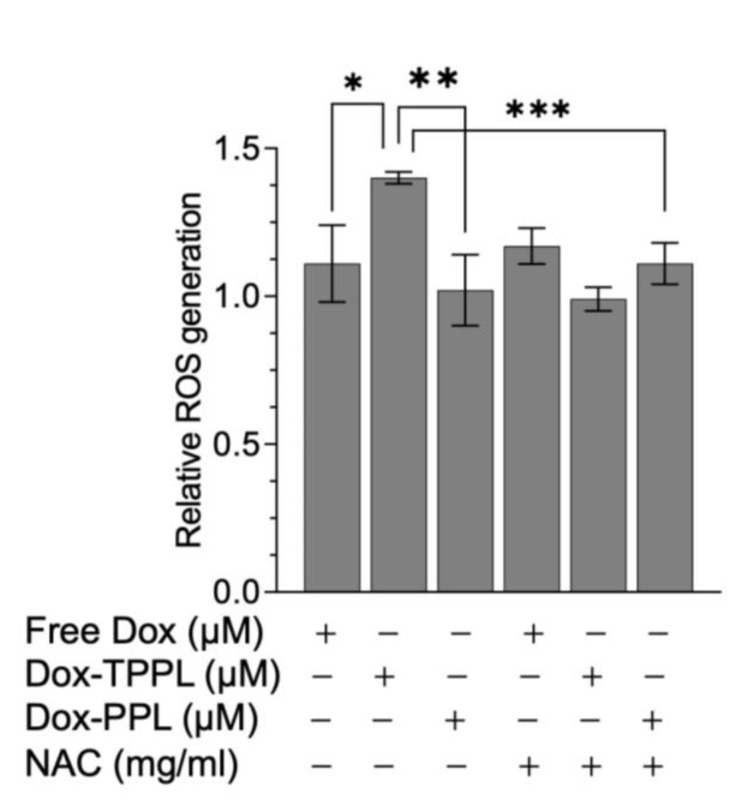
ROS was measured using the H_2_DCFDA assay in HCT116 cells treated with free Dox, Dox-loaded PPLs, and Dox-loaded TPPLs for 24 h and normalized to the corresponding viable cells. Each experiment was performed in triplicate. Data are shown as mean fold change compared to untreated control ± STD. * *p* = 0.011; ** *p* = 0.003; *** *p* = 0.0008.

**Table 1 pharmaceutics-16-00950-t001:** Physicochemical properties of liposomes.

Formulation	Particle Size (nm)	PDI	Zeta Potential (mV)
Empty PPL	117.51 ± 2.82	0.19 ± 0.02	−13.20 ± 0.30
Empty TPPL	135.76 ± 4.29	0.15 ± 0.01	6.07 ± 0.17
Dox-PPL	109.66 ± 2.09	0.20 ± 0.01	−23.63 ± 0.21
Dox-TPPL	126.50 ± 1.29	0.18 ± 0.02	9.97 ± 0.90
RhB-PPL	147.89 ± 1.31	0.15 ± 0.03	−14.97 ± 0.31
RhB-TPPL	157.65 ± 0.18	0.18 ± 0.03	6.80 ± 0.00

## Data Availability

Data are contained within the article.

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
