# Peer review of "Mitochondria-Targeted Liposomes for Drug Delivery to Tumor Mitochondria"

_pharmaceutics, 2024, doi:10.3390/pharmaceutics16070950_

Round 1
Reviewer 1 Report
Comments and Suggestions for Authors
The manuscript investigates the mitochondria-targeted ability of liposomes prepared with DSPE-PEG-TPP polymer.
The novelty of the work must be better specified. DSPE-PEG-TPP is a known and also commercially available polymer. The novelty of the study should rely on its employment to prepare liposomal formulation that can target the drug to the mitochondria.
Why according to the title the prepared liposomes should be pH-sensitive? The aspect has not been adequately discussed and the faster DOX release at pH 5.6 than 7.4 does not justify the liposomes pH-responsiveness. This aspect must be clarified and discussed. Otherwise, “pH-sensitive” must be deleted from the title.
Why the release was faster at pH 5.6 than 7.4? Did drug degradation phenomena occur during DOX release?
Figure 7 and 8 are incorrectly numbered as Figure 4 and Figure 5.
Author Response
Thank you very much for taking the time to review this manuscript. Please find the detailed responses below and the corresponding revisions/corrections in track changes in the re-submitted files.
Comments 1. The manuscript investigates the mitochondria-targeted ability of liposomes prepared with DSPE-PEG-TPP polymer. The novelty of the work must be better specified. DSPE-PEG-TPP is a known and also commercially available polymer. The novelty of the study should rely on its employment to prepare liposomal formulation that can target the drug to the mitochondria.
Response 1. The use of the TPP group conjugated with DSPE-PEG chains on the surface has not yet been displayed for self-assembled liposomal formulations in terms of their stability and pH dependency. In this study, the mitochondria-targeting polymer, DSPE-PEG-TPP polymer conjugate, was successfully synthesized and Dox molecules were encapsulated into these stabilized, mitochondria-targeted liposomes to enhance the therapeutic index of Dox. Then the comparative study was conducted to characterize the physicochemical properties and cytotoxicity of both empty and Dox-loaded TPPL and PPL. These points are included in the discussion section of the revised manuscript.
Comments 2. Why according to the title the prepared liposomes should be pH-sensitive? The aspect has not been adequately discussed and the faster DOX release at pH 5.6 than 7.4 does not justify the liposomes pH-responsiveness. This aspect must be clarified and discussed. Otherwise, “pH-sensitive” must be deleted from the title. Why the release was faster at pH 5.6 than 7.4? Did drug degradation phenomena occur during DOX release?
Response 2. At acidic pH 5.6, the amine and hydroxyl groups on the Dox molecule become protonated, which increases the solubility and hydrophilicity of Dox in an aqueous release medium compared to neutral pH 7.0. This higher solubility allows Dox to more easily diffuse out of the dialysis tubing and into the surrounding environment, leading to a faster release rate. Also, the release studies were designed considering the sink conditions for Dox drug molecules, so the receptor solution was kept at least ten times larger than the solubility of Dox molecules in an aqueous environment, which prevents the precipitation of the released drug molecules in the media.
We have not checked the degradation tendency of Dox molecules via HPLC or LC-MS methods. However, we quantified the released drug amount by considering the maximum absorbance wavelength unique to the drug molecule, which probably eliminates the effect of degradation products, if there is any. pH-sensitive is deleted from the title.
Comments 3. Figure 7 and 8 are incorrectly numbered as Figure 4 and Figure 5.
Response 3. It is corrected.
Reviewer 2 Report
Comments and Suggestions for Authors
The authors developed mitochondria-targeted PEGylated liposomes (TPPL) as nanocarriers to improve the therapeutic efficacy and reduce side effects of chemotherapeutic agents that function in mitochondria. They synthesized a special polymer conjugate and used it to prepare the TPPL formulation. The study compared the TPPL loaded with doxorubicin (Dox-TPPL) to non-targeted PEGylated liposomes (PPL) in terms of physicochemical properties, drug release profile, cellular uptake, mitochondrial localization, and anticancer effects in HCT116 cancer cells.
I have the following suggestions for improvements:
1. The section 2.2: The synthesis of the DSPE-PEG-TPP polymer conjugate is described briefly; however, more detailed information on the reaction conditions (e.g., temperature specifics, reaction time for each step) could be beneficial for reproducibility.
2. The section 2.3: In the section detailing liposome preparation, the authors mention the use of different methods for empty and Dox-loaded liposomes. It would be advantageous to explain why different methods were chosen and if any comparative studies were considered to standardize the methodology.
3. Data representation on Figs 3-8: Some figures, particularly those related to liposome characterization and in vitro experiments, could benefit from more detailed legends explaining what each data point or error bar represents. Including more comprehensive data visualization, such as box plots or scatter plots, could also provide deeper insight.
4. Figures 7 and 8 have wrong numbers.
5. Discussion: A more detailed discussion comparing the results of this study with existing studies, particularly those involving other types of mitochondrial-targeting drug delivery systems is missing. How do the efficacy and safety profiles of the developed liposomes compare with those in published literature?
6. The authors may find helpful a recent paper reporting lipid formulations of anticancer agents for mitochondrial drug delivery: Composition-Switchable Liquid Crystalline Nanostructures as Green Formulations of Curcumin and Fish Oil, ACS Sustainable Chem. Eng. 2021, 9, 44, 14821–14835.
Comments on the Quality of English LanguageMinor editing of English language needed.
Author Response
Thank you very much for taking the time to review this manuscript. Please find the detailed responses below and the corresponding revisions/corrections in track changes in the re-submitted files.
Comments and Suggestions for Authors
The authors developed mitochondria-targeted PEGylated liposomes (TPPL) as nanocarriers to improve the therapeutic efficacy and reduce side effects of chemotherapeutic agents that function in mitochondria. They synthesized a special polymer conjugate and used it to prepare the TPPL formulation. The study compared the TPPL loaded with doxorubicin (Dox-TPPL) to non-targeted PEGylated liposomes (PPL) in terms of physicochemical properties, drug release profile, cellular uptake, mitochondrial localization, and anticancer effects in HCT116 cancer cells.
I have the following suggestions for improvements:
Comments 1. The section 2.2: The synthesis of the DSPE-PEG-TPP polymer conjugate is described briefly; however, more detailed information on the reaction conditions (e.g., temperature specifics, reaction time for each step) could be beneficial for reproducibility.
Response 1: The section 2.2. the synthesis of the DSPE-PEG-TPP polymer conjugate was rewritten and explained in detail in the revised manuscript.
Comments 2: The section 2.3: In the section detailing liposome preparation, the authors mention the use of different methods for empty and Dox-loaded liposomes. It would be advantageous to explain why different methods were chosen and if any comparative studies were considered to standardize the methodology.
Response 2: Both empty and Dox-loaded liposomes were prepared by the same method, which was the thin film hydration method as presented in section 2.3. The formulations of empty and Dox-loaded liposomes were the same; the only difference was the ultracentrifugation step to separate free Dox from the formulations. We made it more clear in the section 2.3. DSPE-PEG-TPP polymer was chosen due to less cytotoxicity to cells. During the standardization process, we used different liposomal formulations, for example, different molar percentages of PEG-PE and DSPE-PEG-TPP, and cross-linking liposomes. The formulations used in this manuscript were chosen due to their physicochemical characteristics and stability.
Comments 3: Data representation on Figs 3-8: Some figures, particularly those related to liposome characterization and in vitro experiments, could benefit from more detailed legends explaining what each data point or error bar represents. Including more comprehensive data visualization, such as box plots or scatter plots, could also provide deeper insight.
Response 3: Figure 3-8 legends are rewritten with detailed explanations. Figures 3 and 5 are reorganized.
Comments 4: Figures 7 and 8 have wrong numbers.
Response 4: Figure legends are corrected.
Comments 5: Discussion: A more detailed discussion comparing the results of this study with existing studies, particularly those involving other types of mitochondrial-targeting drug delivery systems is missing. How do the efficacy and safety profiles of the developed liposomes compare with those in published literature?
Response 5: They are discussed in the discussion section of the revised manuscript.
Comments 6: The authors may find helpful a recent paper reporting lipid formulations of anticancer agents for mitochondrial drug delivery: Composition-Switchable Liquid Crystalline Nanostructures as Green Formulations of Curcumin and Fish Oil, ACS Sustainable Chem. Eng. 2021, 9, 44, 14821–14835.
Response 6: Thank you for suggesting this very nice and interesting systematic study about the comparison of different formulations to obtain self-assembled lipidic particles. We preferred to demonstrate only one composition for our lipidic nanoparticle system and focus on its biological functioning, rather than a comparative representation like in this literature.
Reviewer 3 Report
Comments and Suggestions for Authors
The authors show how liposomes formulated with triphenylphosphonium cation can target Doxorubicin to the mitochondria. The targeting should be stronger to cancer cells since their mitochondria have a more negative membrane potential. The experiments sound well done and good characterization data are presented. There are two points for revision:
(1) line 183, please change ‘distorted’ to ‘disrupted’
(2) can some ideas be provided on why the zeta potential changes upon encapsulation of Dox as given in table 1?
Author Response
Thank you very much for taking the time to review this manuscript. Please find the detailed responses below and the corresponding revisions/corrections in track changes in the re-submitted files.
Comments and Suggestions for Authors
The authors show how liposomes formulated with triphenylphosphonium cation can target Doxorubicin to the mitochondria. The targeting should be stronger to cancer cells since their mitochondria have a more negative membrane potential. The experiments sound well done and good characterization data are presented. There are two points for revision:
Comments 1: line 183, please change ‘distorted’ to ‘disrupted’.
Response 1: It is changed.
Comments 2: can some ideas be provided on why the zeta potential changes upon encapsulation of Dox as given in table 1?
Response 2: Doxorubicin (Dox) is a positively charged molecule due to the presence of an amino group. PEGylated liposome (PPL) has a negative surface charge. When Dox is encapsulated in PPL, the positively charged Dox molecules adsorb onto the negatively charged surface of the PPL. This can partially neutralize the negative surface charge and lead to a more negative zeta potential of Dox-loaded PPL compared to the empty-PPL (Table 1). Both Dox and DSPE-PEG-TPP groups carry a positive charge. When Dox is encapsulated in TPPL, the positively charged Dox molecules adsorb onto the positively charged surface of the TPPL. The positive charges from both Dox and TPP accumulate on the liposome surface, leading to an overall more positive zeta potential for Dox-loaded TPPL compared to empty TPPL. The positive charge from the TPP groups alone in empty TPPL determines the zeta potential, which is less positive than the Dox-loaded TPPL (Table 1). It is added to the results section 3.2 of the revised manuscript.
Round 2
Reviewer 1 Report
Comments and Suggestions for Authors
The authors have addressed the reviewer comments.